# Identifying Barriers and Pathways Linking Fish and Seafood to Food Security in Inuit Nunangat: A Scoping Review

**DOI:** 10.3390/ijerph20032629

**Published:** 2023-02-01

**Authors:** Meghan Brockington, Dorothy Beale, Josephine Gaupholm, Angus Naylor, Tiff-Annie Kenny, Mélanie Lemire, Marianne Falardeau, Philip Loring, Jane Parmley, Matthew Little

**Affiliations:** 1Department of Population Medicine, University of Guelph, Guelph, ON N1G 2W1, Canada; 2School of Public Health and Social Policy, University of Victoria, Victoria, BC V8P 5C2, Canada; 3Centre de Recherche CHU de Québec, Université Laval, Quebec, QC G1V 0A6, Canada; 4Département de Médecine Sociale et Préventive, Université Laval, Quebec, QC G1V 5C3, Canada; 5Institut de Biologie Intégrative et des Systèmes, Université Laval, Quebec, QC G1V 0A6, Canada; 6Centre D’Études Nordiques (CEN), Université Laval, Quebec, QC G1V 0A6, Canada; 7Department de Biologie, Université Laval, Quebec, QC G1V 0A6, Canada

**Keywords:** food security, fish, seafood, fisheries, Arctic, Arctic Canada, food sovereignty, wildlife management

## Abstract

Background: Fish and seafood play an important role in improving food security in Inuit Nunangat. Therefore, this scoping review aims to explore (1) what topics and/or themes have been widely explored in the literature related to barriers and pathways linking fish and seafood to food security; (2) where research, policy, and action gaps exist; and (3) how fisheries currently contribute to food security. Methods: A systematic search of peer-reviewed articles was conducted using six databases. Articles were screened by two independent reviewers. Eligible studies included primary research conducted in Inuit Nunangat that explored the roles of fish and seafood in food security. Results: Thirty-one articles were included for review. Overall, we found that fisheries can influence food security through direct pathways (e.g., consuming fish for nutrition), and through indirect pathways such as increasing household purchasing power (e.g., through employment). Research indicated that policies relating to wildlife and fisheries management need to be integrated with food and health policies to better address food insecurity in Inuit Nunangat. Conclusion: Future research is needed to establish a more robust understanding of the explicit mechanisms that fish and seafood harvest and/or the participation in commercial fisheries alleviates household food insecurity.

## 1. Introduction

For millennia, the harvest, sharing and consumption of country foods, including berries, marine and terrestrial mammals, seafood and fish, have had important health, psychosocial, and cultural benefits for Inuit residing in Inuit Nunangat (the Inuit homelands in Canada) [1,2,3]. However, colonial processes (e.g., environmental dispossession and loss of traditional knowledge) and current government policies (e.g., hunting restrictions and quotas) have contributed to decreased consumption of country food and increased reliance on calorically dense, low-nutrient market food [4,5,6,7,8]. Exacerbated by climate and socioeconomic change [9,10], this dietary transition has resulted in increased incidence of diet-related diseases [8,11], as well as persistently high prevalence of food insecurity in Inuit communities relative to the national average [12,13,14].

In Canada, according to the 2017 Aboriginal Peoples Survey, 76% of Inuit over the age of 15 and living in Inuit Nunangat report experiencing food insecurity [15]. The prevalence of food insecurity was highest in Nunavut (77.6%), followed closely by Nunavik (77.3%), Inuvialuit Settlement Region (ISR) (68.5%) and Nunatsiavut (68.4%) [15]. These prevalence estimates are eight to nine times higher than the Canadian average of 8.8% [15].

It is widely accepted that country foods are important sources of dietary micro and macro nutrients [16,17,18,19]. In particular, marine fish and seafood (including anadromous Arctic Char, Greenland Cod, Capelin, blue mussels, and other species) are important sources of protein, essential fatty acids, vitamins D, A and B, and minerals (e.g., calcium, phosphorus, iodine, zinc, iron, and selenium) [20]. Additionally, activities associated with country food harvest preparation and sharing are fundamental to the cultural identity of Inuit communities [21]. For these nutritional, cultural, and wellness reasons, country foods are a key pillar of food security and sovereignty for Indigenous communities in Canada [22,23,24,25].

In northern Canada [26,27,28,29,30,31,32] and internationally [33], there is strong evidence that fisheries can play a significant role in food security. While fisheries are understood differently in various contexts, we define fisheries as the enterprise of harvesting anadromous fish, crustaceans, and/or molluscs (not marine mammals) for subsistence and/or commercial purposes. Sustainable, Inuit-managed commercial fisheries could be a vehicle for increased traditional food production, food sovereignty, and community economic development [26,27,28,31,32]. It is suggested that in parts of northern and Arctic Canada, annual harvestable fish supply exceeds the actual level of fish harvested [26,33]. Therefore, considering the potential for fisheries to contribute to enhancing food security, this review aims to explore the range, nature, and extent of literature examining the role of fish in food security in Inuit Nunangat. We will begin by providing a contextual overview of fisheries by region in Inuit Nunangat before describing the methods and synthesizing the literature identified by our review.

For the purposes of this review, we use the Food and Agricultural Organization of the United Nations’ definition of food security as adopted by the Nunavut Food Security Coalition: “Food security exists when all people at all times, have physical and economic access to sufficient, safe, and nutritious food to meet their dietary needs and food preferences for an active and healthy life”. From this definition, the Nunavut Food Security Coalition outlines four components of food security: (1) Availability (enough wildlife on the land and groceries in the store), (2) Accessibility (adequate money for hunting equipment or store-bought food, and the ability to obtain it), (3) Quality (healthy food that is culturally valued), and (4) Use (knowledge about how to obtain, store, prepare, and consume food). In addition to food security, this review adopts a food sovereignty approach which recognizes the rights of Inuit to define their own hunting, fishing, land and water policies and governance. It also acknowledges the rights of Inuit to determine the future of their food systems [34].

## 2. Context: Fisheries Overview by Region

Before delving into the literature review, we provide an overview of the commercial fisheries context in the different regions of Inuit Nunangat: Nunavut, the Inuvialuit Settlement Region, Nunavik, and Nunatsiavut. Under important land claims agreements, Inuit have experienced changes in sovereignty and involvement in decision-making compared to the initial regional settlement periods [35]. Within each land claim agreement, co-management boards exist [35]. These boards are responsible for the governance of fisheries resources in conjunction with Fisheries and Oceans Canada (DFO) and the respective provincial, territorial governments, as well as regional and local organizations and stakeholders, whose decisions and activities must conform to the rules and regulations established in the 2019 amendment of the federal *Fisheries Act* [35,36]. For most of these fisheries, co-management boards ensure that sections of the land claims agreements relevant to fisheries are implemented and establish the opportunity for dialogue between all stakeholders that is then communicated to the Minister of Fisheries and Oceans.

### 2.1. Nunavut

Nunavut’s well-established commercial fishing industry began in the 1960’s. Nunavut has total allocation quota for Turbot of 11,500 metric tonnes, 10,995 metric tonnes for Shrimp, and 362,873 kg for Arctic Char [37]. The main species caught in the territory are Turbot, Northern Shrimp, Lake Trout, Arctic Char, and Arctic Grayling, with the approximate commercial value of Turbot, Shrimp, and Arctic Char at CAN$86.3 million in 2015 [37]. Cambridge Bay in the Kitikmeot region and Pangnirtung in the Qikiqtani region are host to two of the largest commercial Arctic Char fisheries in Arctic Canada [38]. The Qikiqtani region (encompassing a large region and 13 communities from Grise Fjord in the High Arctic to Sanikiluaq on the Belcher Islands) is also the territory’s leader in offshore Turbot and Shrimp commercial harvests [39]. Four large-scale commercial companies operate in this region in Baffin Bay, Davis Strait, and Hudson Strait [40]. The territory also has five processing plants for Arctic Char, Turbot, Shrimp, and whitefish. In Nunavut, fisheries are co-managed in conformity with the Nunavut Land Claims Agreement and DFO’s *Fisheries Act*. Different organizations are involved in co-management including the Nunavut Wildlife Management Board, which is the main regulator of wildlife-related issues in the region and is responsible for communicating decisions made with co-managing partners including DFO, who can amend or reject decisions in case of conservation concern [39,40,41]. Local Hunters and Trappers Organizations (HTOs) also have a key role to play in co-management, as well as other actors including fish plants and commercial fishers [40].

### 2.2. Inuvialuit Settlement Region

In the Inuvialuit Settlement Region of the Northwest Territories (NWT), the Fisheries Joint Management Committee works collaboratively with DFO to co-manage fishery and marine mammal resources. Further, the Inuvialuit Final Agreement affirms the right of Inuvialuit (in both the Yukon and NWT), and co-operatives to be issued non-transferable fishing licences to fish in any Inuvialuit Settlement Region waters, while the Beaufort Sea Integrated Fisheries Management Framework protects the sustainability and health of the resources, including key traditional species, such as Arctic Char. In summary, DFO, the Inuvialuit Regional Corporation, the Inuvialuit Game Council, and the Fisheries Joint Management Committee co-manage the resources and oversee the region’s sustainable development [42]. However, to date, commercial fishing in the NWT has been limited to Great Slave Lake, where the population is primarily non-Inuit harvesting freshwater species such as Lake Whitefish, Inconnu, Northern Pike, and Walleye [39,43]. In 2010, fish harvest in the NWT had an estimated value of CAN$791,000 with a volume of 525,000 kg [43,44].

### 2.3. Nunavik

Nunavik has an important and growing Shrimp industry in the Hudson and Davis Straits that was initiated in 1978 and currently operates via two vessels [45]. Makivik was then created as part of the James Bay and Northern Quebec Agreement, with an initial role of managing the land claim funds. Makivik’s role grew through the years into a central mandate representing Nunavik Inuit on their rights as related to social and economic development, infrastructure, and protection of Inuit culture, language, and lands [46]. Makivik Corporation is the full owner of one fishing licence and shares one with the Qikiqtaaluk Corporation of Nunavut, both of which are used by Newfound Resources Ltd., Makivik’s main fishing partner. Over the years, royalties from the fishery venture, which in 2017–2018 totalled CAN$7.2 million, have been invested back into the communities of Nunavik in the form of infrastructure, training, wages and in Shrimp research [45,47]. In terms of co-management, the Nunavik Marine Region Wildlife Board (NMRWB) manages the area’s marine resources through consultation with Nunavimmiut communities’ local and regional wildlife management organisations, and with oversight, approval and veto power vested in the federal Minister of DFO [39]. It was created under the Nunavik Inuit Land Claims Agreement in 2008. The NMRWB is composed of three Makivik Corporation members, a representative from DFO, a representative from the Ministry of the Environment and Climate Change, one representative from the Government of Nunavut, and a chairperson. Nunavik also has important populations of Arctic Char, but no formal commercial fishery of that species is currently in place. However, some commercial fishing activities have occurred, including via the Hudson Bay Company as early as in 1880, and some exploratory commercial activities in more recent decades. Now, individual fishers can own exploratory commercial fishing licences for Arctic Char, which are produced by the Quebec *Ministère des Forêts, de la Faune et des Parcs (MFFP)*. Indeed, while Arctic Char management is ensured by DFO in most northern territories, it is the provincial government that oversees the conservation and management of Arctic Char in Quebec [48].

### 2.4. Nunatsiavut

The 2005 Labrador Inuit Land Claims Act governs fishery resources in this region, with the Torngat Joint Fisheries Board, an Indigenous cooperative formed as part of the land claim, acting as the primary advisory board to the Ministry of Fisheries and Oceans Canada on the management of fishery resources. The Board is composed of six appointed members—two from Government of Canada, one from the Government of Newfoundland and Labrador, and three from the Nunatsiavut Government—in addition to a single independent chairperson [36]. Snow crab, Shrimp and Turbot are the dominant commercially harvested species in Nunatsiavut, with a direct value from sales of CAN$825,312 for Turbot and CAN$14,505,827 for Shrimp, and a harvest value of CAN$1.48 million for snow crab in 2017 [36,49].

## 3. Materials and Methods

Due to the broad guiding research question and the multidisciplinary nature of literature on fish, seafood, and fisheries as it relates to food security, sovereignty and nutrition, a scoping review methodology was used to explore key concepts that arise from diverse, complex, and broad research [50]. This review follows the framework established by Arksey and O’Malley [51] and later updated by Levac and colleagues [52]. In summary, we undertook the following steps: (1) identify research questions; (2) identify relevant studies; (3) study selection; (4) chart the data; and (5) collate, summarize, and report results. Methods and findings were reported according to the Preferred Reporting Items for Systematic reviews and Meta-Analyses (PRISMA) extension for Scoping Reviews checklist [53].

### 3.1. Stage 1: Identifying the Research Question

Literature that examines the role of fish, seafood, and fisheries in food security in Inuit Nunangat is derived from several fields of research. Thus, our overall guiding research questions are: (1) what topics and/or themes have been widely explored in the literature related to barriers and pathways linking fish and seafood to food security; (2) where do research, policy, and action gaps exist; and (3) how do fisheries currently contribute to food security?

### 3.2. Stage 2: Identifying Relevant Studies

In consultation with a research librarian at the University of Guelph, we conducted a systematic search using Medline via PubMed, GEOBASE, Web of Science, Cab Direct, AGRICOLA, and ProQuest. During the development of the protocol, key concepts and expanded search terms were identified. Table 1 provides an overview of the search terms that informed our search and a complete search strategy for each database is available in the Appendix A (Table A1, Table A2 and Table A3). In addition to the structured database search, a hand search was conducted using the reference lists of included articles and Google Scholar to identify relevant articles that met the inclusion criteria.

### 3.3. Stage 3: Study Selection and Eligibility Criteria

All peer-reviewed literature available in English from any publication date was eligible for inclusion. Further, literature was eligible for inclusion if there was at minimum a partial focus on the role of anadromous fish, crustaceans, molluscs, and fisheries in food security in the Arctic. Studies that strictly investigated marine and freshwater mammals were ineligible for inclusion. We anticipated the need to use proxy terms for “food security” due to the variety of study designs and measures in the literature. Therefore, studies that included the following terms were eligible for inclusion: food supply, nutrition security, food system, food sovereignty, food availability, food accessibility, and food utilization.

The target population for this scoping review was Inuit living in one of Inuit Nunangat’s four land claims regions: Nunavut, Nunavik (Northern Quebec), Nunatsiavut (Newfoundland and Labrador), and the Inuvialuit Settlement Region (Northwest Territories and the Yukon). Studies were ineligible if they focused exclusively on peoples living in Canada of non-Inuit descent. The rationale for the participant inclusion criteria is that prevalence of food insecurity differs by ethnicity and race [4].

Databases were accessed initially from March to June 2021. An additional hand search was conducted in June 2022. Based on the authors’ predetermined eligibility criteria, two independent reviewers (MB and JG) screened the articles in a two-step process using the review management program DistillerSR. During level one screening, titles and abstracts were screened. Articles proceeded to level two screening if both the primary and secondary reviewers decided the inclusion criteria were met, or if both reviewers were unclear if the inclusion criteria were met. During the level two screening, the full text articles were reviewed against the inclusion and exclusion criteria. Disagreements between reviewers were solved through discussion and consensus for both level one and level two. The Kappa statistics, refers to the degree of agreement between reviewers, for level one and level two screening were 0.79 (moderate agreement) and 0.86 (strong agreement), respectively [56].

### 3.4. Stage 4: Charting the Data

Data charting was conducted using a tool specifically developed for this scoping review. The charting process was piloted by two reviewers (MB and DB) on 10 articles to ensure consistency in results. Subsequently, one reviewer (DB), independently performed the data extraction on the remainder of the included articles. For each article, we extracted data on the author(s), date of publication and journal, geographical focus, study methodology, aim of research, and main conclusions related to the research questions.

### 3.5. Stage 5: Collating, Summarizing, and Reporting the Research

Due to the broad range of literature captured in this review, an inductive thematic analysis was conducted in Nvivo 12 to provide a coherent synthesis of the current state of knowledge [57]. A thematic analysis is a method for identifying and reporting patterns (or themes) in a dataset to provide rich descriptive detail [57]. Text from the whole articles was reviewed, coded using an inductive method, and these codes were iteratively refined and grouped into themes and presented in the results [57].

## 4. Results

A total of 31 articles were identified for in-depth review (Figure 1).

## 5. Study Characteristics

### 5.1. Study Design

A summary of study characteristics can be found in Table 2 and a detailed overview can be found in the Appendix A Materials. Of the 31 articles included in this review, 39% (12 articles) used qualitative methods [58,59,60,61,62,63,64,65,66,67,68,69], 52% (16 articles) used quantitative methods [11,16,21,34,59,60,61,62,63,64,65,66,67,68,69,70,71], and 9% (3 articles) used “other” methods [70,71,72]. Specific methods used in the studies included semi-structured interviews (10 articles) [58,59,60,61,64,65,66,67,69,71], cross-sectional surveys (11 articles) [11,16,21,69,73,74,75,76,77,78,79,80], quantitative databases for contaminants, fish stocks, or harvesting data (seven articles) [34,59,61,62,70], focus groups (three articles) [61,62,66], descriptive/narrative (three articles) [63,70,72], and open-answer interviews (one article) [68]. Twelve articles quantified and characterized food intake using either a food frequency questionnaire and/or 24-h dietary recalls [11,16,21,73,74,75,76,77,78,79,80].

### 5.2. Geographic Focus

Nunavut was the most studied region (eleven articles) [11,59,61,62,65,67,69,70,71,72,81], followed by Nunavik (nine articles) [58,63,64,71,74,75,76,77,79]. One article focused on Nunatsiavut [66]. Eleven studies focused on multiple Arctic regions [21,33,60,73,78,80,82,83,84,85], both nationally and globally [21,33,60,73,78,80,82,83,84,85]. No articles focused exclusively on the Inuvialuit Settlement Region.

**Table 2 ijerph-20-02629-t002:** Characteristics, pathways, and themes identified in the thematic analysis of the 31 articles included in this review examining the role of the fish, seafood, and fisheries in food security in Inuit Nunangat. Articles were ordered by region (Nunavik, Nunavut, ISR, and Nunatsiavut, or multiple regions in Inuit Nunangat for studies covering more than one region).

Author(s)	Year	Location(s): Communities (Regions)	Overview of Study Design	Pathway to Food Security	Themes that Emerged from the Thematic Analysis
Berkes, E. [83]	2003	93 communities (NL, QC, ON, MB, SK, BC, NWT)	Quantitative: Synthesis of harvest study data	Direct; indirect	Subsistence fisheries and household fish access; increasing self-determination through fisheries
Falardeau, M., et al. [69]	2022	Cambridge Bay and Kitikmeot marine region (NU)	Mixed Method: Biophysical data and semi structured interviews		Climate change
Kenny, TA., et al. [16]	2018	36 communities (ISR, NU, NL) sampled for the 2007–08 Inuit Health Survey	Quantitative: Cross-sectional survey	Direct	Fish as a source of micronutrients and macronutrients; fish consumption and links to health
Laird, B D. et al. [80]	2013	36 communities (ISR, NU, NL) sampled for the 2007–08 Inuit Health Survey	Quantitative: Cross-sectional survey	Direct	Fish as a source of micronutrients and macronutrients; fish consumption and links to health
Lambden, J., Receveur, O., Kuhnlein, H.V. [68]	2007	Multiple Arctic Canadian communities	Qualitative: Open-ended questions	Direct	Fish and wellbeing; climate change
Rosol, R., et al. [21]	2016	36 communities (ISR, NU, NL) sampled for the 2007–08 Inuit Health Survey	Quantitative: Cross-sectional survey	Direct; indirect	Fish as a source of micronutrients and macronutrients; increasing self-determination through fisheries
Tai, T. C., et al. [85]	2019	Arctic Canada	Quantitative: Catch data; integrated modelling		Sustainability and stability of fisheries and fish-related community programs
Walker, E. [73]	2020	Aklavik (NWT), Fort McPherson (NWT), Old Crow (YK)	Quantitative: cross-sectional study	Direct	Fish consumption and links to health
Watts, P., et al. [82]	2017	Various: 56 communities in Nunavik (QC), ISR (NWT), and Hudson Bay coastline (QC)	Quantitative	Direct	Fish as a source of micronutrients and macronutrients; subsistence fisheries and household fish access; climate change
Zeller, D., et al. [33]	2011	Arctic Canada, Russia, and USA	Quantitative	Direct; indirect	Subsistence fisheries and household fish access; climate change; sustainability and stability of fisheries and fish-related community programs
Blanchet, C., et al. [75]	2000	Nunavik (QC)	Quantitative: cross-sectional study	Direct	Fish as a source of micronutrients and macronutrients
Dewailly, E., et al. [74]	2003	Southern Québec, James Bay (QC), Nunavik (QC)	Quantitative: Cross-sectional study	Direct	Fish as a source of micronutrients and macronutrients; fish consumption and links to health
Gagné, D., et al. [79]	2012	10 communities in Nunavik (QC)	Quantitative: Cross-sectional study	Direct	Fish as a source of micronutrients and macronutrients
Gautier, L., et al. [58]	2016	Nunavik (QC)	Qualitative	Direct	Fish and wellbeing; sustainability and stability of fisheries and fish-related community programs
Gombay, M. [63]	2006	Nunavik (QC)	Review and commentary primary qualitative data	Direct	Fish and wellbeing; socio-economic change
Lemire, M., et al. [76]	2015	14 communities in Nunavik (QC) sampled for the 2004 Qanuipitaa? Health Survey	Quantitative: Cross-sectional survey	Direct	Fish consumption and links to health
Lucas, M., et al. [77]	2010	14 communities in Nunavik (QC) sampled for the 2004 Qanuipitaa? Health Survey	Quantitative Cross-sectional survey	Direct	Fish as a source of micronutrients and macronutrients
Pellerin, J., & Grondin, J. [71]	1998	Ungava Bay, Nunavik (QC)	Mixed methods: Fish sampling and Inuit stakeholder interviews	Direct	Fish consumption and links to health; climate change
Rapinski, M., et al. [64]	2018	Kangiqsujuaq and Ivujivik, Nunavik (QC)	Qualitative: semi-structured interviews	Direct	Fish as a source of micronutrients and macronutrients
Galappiththi, E. K., et al. [61]	2019	Pangnirtung (NU)	Qualitative: participant observations, semi-structured interviews, focus group	Direct; indirect	Fish and wellbeing; increasing household purchasing power; increasing self-determination through fisheries; climate change; socio-economic change
Galappathth, E. K., et al. [60]	2021	Pangnirtung (NU) and Kunjankalkulam, Sri Lanka	Qualitative: Comparative case study	Direct; indirect	Fish and wellbeing; increasing self-determination through fisheries; climate change
Gilbert, S. Z., et al. [65]	2020	Cambridge Bay and Pond Inlet (NU)	Qualitative: semi-Structured interviews	Direct	Fish and wellbeing; climate change; socio-economic change
Hu, X. F., et al. [11]	2018	25 communities (NU) sampled for the 2007–08 Inuit Health Survey	Quantitative: Cross-sectional survey	Direct	Fish as a source of micronutrients and macronutrients
Lysenko, D., & Schott, S. [81]	2019	18 communities (NU)	Quantitative	Indirect	Increasing self-determination through fisheries; increasing self-determination through fisheries; socio-economic change
Nancarrow, T., & Chan, H. M. [62]	2003	Repulse Bay (NU) and Kugaaruk (NU)	Qualitative: focus groups		Climate change
Pannikar, B., & Lemmond, B. [59]	2020	Cambridge Bay (NU) and Kugluktuk (NU)	Qualitative: semi-structured interviews	Direct; indirect	Fish and wellbeing; increasing self-determination through fisheries; climate change; socio-economic change
Schott, S., et al. [70]	2020	Gjoa Haven (NU)	Commentary	Indirect	Increasing self-determination through fisheries
Stecyk, K. [72]	2018	Nunavut	Commentary	Indirect	Increasing household purchasing power; increasing self-determination through fisheries
Kuhnlein, H., & Receveur, O. [86]	2007	Yukon and NWT	Quantitative: Cross sectional	Direct	Fish as a source of micronutrients and macronutrients
Kourantidou, M., Hoagland, P., & Bailey, M. [66]	2022	Nain and Makkovik, Nunatsiavut (NL)	Qualitative: Focus groups semi-structured interviews	Indirect	Increasing household purchasing power; increasing self-determination through fisheries
Ford, J. D., Smit, N., & Wandel, J. [9]	200	Arctic Bay (NU)	Qualitative: Semi structured interviews		Climate change
Chan, H. M. [84]	1998	N/A	Literature review and database compilation	Direct	Fish consumption and links to health

## 6. Barriers and Pathways Linking Fisheries to Food Security

The thematic analysis revealed four direct, two indirect pathways, and three barriers linking fish to food security in Inuit Nunangat (see Table 2 for references sourced for each pathway and theme).

### 6.1. Direct Pathways to Food Security through Fish Consumption

#### 6.1.1. Fish and Well-Being

As defined by the FAO and Nunavut Food Security Coalition, Inuit food security is dependent on cultural values, skills and spirituality that is learned through the harvest, sharing, and preparing of country foods. Qualitative studies consistently show that fish consumption is seen as healthy, tasty, and important to cultural identity across Inuit Nunangat [58,65,68].

Intra- and inter-community food and knowledge sharing networks are an important aspect of community-level food security. In a qualitative study conducted in Pangnirtung, Nunavut, Galappaththi and colleagues (2019) noted that some types of Inuit knowledge are becoming lost intergenerationally, such as survival skills on ice, reading the sky, sewing seal skin and handling dog teams [61]. However, generational knowledge transfer of traditional skills and knowledge remains an essential way to adapt to changes in the Arctic [60,61]. In the same study, 84% of Inuit fishers interviewed indicated that they had learned about fishing from Elders and/or extended family members [61]. Further, Inuit fishers unanimously agreed that learning from Elders as a youth was critical for adapting to a changing Arctic. In addition to knowledge sharing, food sharing networks are deeply connected to food security. For instance, food sharing networks increase access to harvested char. Fishers will often share their catch with relatives and Elders, especially sharing with those who cannot fish and hunt for themselves [58,59,61,63]. Other strategies for increasing food security through food sharing include fishers going on local radio to share that they have extra catch and community initiatives that organize food sharing events and programs (such as a soup kitchen) [58].

#### 6.1.2. Fish as a Source of Micronutrients and Macronutrients

Studies included in our review supported well-established evidence on the high nutritional value of fish, including their abundance in micro- and macronutrients, including protein, n-3 polyunsaturated fatty acids, vitamins D, A and B, and minerals (calcium, phosphorus, iodine, zinc, iron, and selenium). It is clear from the literature that Arctic Char, specifically, continues to be central to Inuit culture across the Inuit Nunangat, a dietary staple in subsistence diets and good source of dietary nutrients, even when consumed infrequently [64,77,78,79,80]. For example, in the Kivaliq region of Nunavut, fish only contributes 2% of dietary energy, but 19% of total vitamin D intake [21]. Similarly, in the ISR, fish contributes to 6% of dietary energy, but 17% to protein and 36% to vitamin D requirements [82]. Literature included in our review showed that individuals who reported higher intakes of fish and seafood had higher biomarker concentrations of n-3 polyunsaturated fatty acids, vitamin D, vitamin A, vitamin B-6, and selenium, among other nutrients [11,74,75,77,78,79]. Based on calculations of the potential caloric value of total marine harvests, fish provide a substantial contribution to food security and may contribute all the protein requirements for early childhood nutrition. The high protein value of fish (60–90% of fish total body weight) further underscores its importance to protein intake, a key aspect of food security [82].

#### 6.1.3. Fish Consumption and Links to Health

Fish consumption is associated mostly with several health benefits, as well as some very moderate health risks. There is evidence that fish consumption is protective against many health outcomes, including cardiovascular disease, hypertension, and stroke, likely due to their abundance in n-3 polyunsaturated fatty acids (e.g., eicosapentaenoic acid and docosahexaenoic acid) [64,69]. Despite high levels of smoking and obesity in Inuit communities of Nunavik, as of 1996, Nunavimmiut populations with high fish consumption had a lower low-density-lipoprotein and high-density-lipoprotein ratio, cholesterol, triglyceride, systolic and diastolic blood pressure, and insulin when compared to other Quebecers and James Bay Cree, indicating lower cardiovascular disease risk [74].

Harmful contaminants such as methylmercury (MeHg) and PCBs (polychlorinated biphenyls) remain a concern for Inuit living in Inuit Nunangat [84]. Lemire and colleagues (2015), found that while most country food species were found to have low concentrations of MeHg (0.2–0.5 μg/g), Lake Trout was found to have a very high concentration of MeHg (≥1.0 μg/g), and Northern pike and walleye had a high concentration of MeHg (0.5–1.0 μg/g). Perhaps most importantly, however, Arctic Char did not contribute significantly to MeHg intake, but was an exceptional source of long chain n-3 polyunsaturated fatty acid intakes [76]. Correspondently, Walker et al., (2020) found that hair-MeHg concentration increased with total fish (including inconnu, Broad Whitefish and Dolly Varden) and whale consumption among Gwitchin and Inuvialuit residents of the Western Canadian Arctic, however, these concentrations remained well below Health Canada’s threshold for unsafe exposure [73]. It is also important to consider that fish and whale consumption were grouped together, thus whale consumption may be dominate contributing factor to MeHg intake (as fish typically have low Hg levels). Laird and colleagues (2013) similarly reported that though Arctic Char was consumed often, it only contributed to 8.4% of overall MeHg [69] exposure [80]. Therefore, when compared with other country foods, Arctic Char is not heavily contaminated and exhibits concentrations of Hg well within the acceptable range for human consumption [71,84]. Indeed, this review revealed that substituting country foods high in Hg (e.g., beluga meat, older Lake Trout, seal liver) with Arctic Char would reduce Hg intake without adversely affecting nutrient intake [73,76,77,87].

#### 6.1.4. Subsistence Fisheries and Household Fish Access

Several studies acknowledged the significance of subsistence fisheries to food security [33,82,83]. However, this topic has received little direct attention by researchers in Inuit Nunangat. A few articles provided estimates of the total marine and fish harvest in Inuit Nunangat. Watts et al., (2017) estimated the total marine harvest was 4882 tonnes in the year 2000, with 32% of that being fish (mostly Arctic Char) [82]. Commercial fisheries accounted for only 6% of this total [82]. Berkes (2003) estimated that small scale catches provide harvesters with 42 kg of edible weight per person annually, which is six times higher than the Canadian average [83]. This underscores the importance of the small catch component of fisheries and the subsistence economy for food security in Inuit Nunangat [33,82]. Despite this, there is a dearth of knowledge regarding the relative contributions of small-scale fisheries to community and household food security, distribution of subsistence catches across communities, as well as the importance of the broader subsistence economy in promoting food security across Inuit Nunangat.

### 6.2. Indirect Pathways Linking Fish and Seafood to Food Security

#### 6.2.1. Increasing Household Purchasing Power

Commercial fisheries can harness the economic potential of fish harvesting to contribute to food security by increasing household purchasing power, and in turn their ability to purchase healthy food. Several studies acknowledged the positive impacts of commercial fisheries, small-scale fisheries, or aquaculture on income and community adaptive capacity to socio-economic disruptions and climate change [61,66,72]. In Pangnirtung, Nunavut, Arctic Char and Turbot are co-existing fisheries that contribute to the community’s adaptive capacity by increasing purchasing power and availability of Arctic Char [61]. In addition to the employment opportunities in the commercial fishery operation, the community fish processing plant is an important employer [61]. Similarly, in Nunatsiavut, focus group participants identified that they gain significant non-market economic benefits from participating directly in commercial fishing or indirectly in the subsequent processing of harvested fish [66]. These benefits include bequest value (i.e., the value placed on the opportunity to fish in the future), and the existence value (i.e., the non-use value of the existence of fish in the future) [66].

While no studies have explored whether households or communities involved in the commercial fishing industry experienced increased food security, the role of aquaculture to enhance food security was discussed by Stecyk (2018) [72]. In addition to the economic benefits, year-round aquaculture can improve local access and availability of fish or seafood, while reducing reliance on imported fish [72]. Stecyk (2018) also argues that the Government of Canada should invest in organizations (e.g., Baffin Fisheries), initiatives (e.g., the Commercial Fisheries Freight Subsidy in Nunavut), and research that contributes to growing the fishing industry in the Arctic [72]. Further, they argue that the government should work collaboratively with Indigenous organizations to ensure that the needs of the local communities are being met in the emerging commercial fishing and aquaculture industries [72].

#### 6.2.2. Increasing Self-Determination through Fisheries

Several studies recognized and reiterated the importance of co-management of resources to enhance agency and self-determination [60,61,70,72,81]. Specifically, the decentralization and sharing of fisheries management and governance was cited as an opportunity to realize the right of Inuit to pursue their own food sovereignty and improve food security [59,60,61,83].

Three ideas to enhance bottom-up governance and dismantle inequitable power structures within fisheries management emerged from the literature. First, interviews with community members in Nunatsiavut suggested that fisheries management decisions are driven largely by the economic objectives of the federal government, which often differ from the objectives of communities [80] can lead to the centralization of efforts on species with high commercial value, without considering species that are important to local communities [66]. It can further undermine the agency and self-determination communities have over their fisheries resources and thus their ability to address regional food insecurity [66]. It has therefore been suggested that the development of a co-management agency that includes all Inuit regions, researchers, and decision-makers could serve as a bridge between the multiple levels of government, research, policy, community initiatives and knowledge [21]. This agency could assist in reducing the challenges imposed by government and their policies, by ensuring that food security initiatives were driven by the rights and needs of local communities [21].

Second, articles stressed that political and funding cycles make it difficult to build relationships or enhance governance structures. An important recommendation was that agencies (e.g., DFO and funding agencies) need to adopt a long-term view of fisheries co-management by ensuring continual access to base funding and establishing long-term mandates that are resilient to political leadership change [61,70]. Finally, some authors identified a tension between health and wildlife policies. In general, food security falls under the jurisdiction of public health agencies, while subsistence and commercial fisheries are often overseen by co-management boards (that include DFO). This division of roles and responsibilities enforces barriers to addressing food insecurity through fisheries, such as increasing access to diverse fish and seafood species. Food security and wildlife management policies should be part of an integrated policy framework, where food security issues are balanced with environmental concerns, and the right to self-determination over food systems to meet the needs of individuals and communities [12]. This would require better coordination of the multiple levels of government, researchers, and community agencies [70,81].

### 6.3. Other Considerations to Fish-Related Food Security

Articles identified several barriers to leveraging fisheries to support food security in Inuit Nunangat, including climate change, socio-economic changes, and fisheries sustainability and stability.

#### 6.3.1. Climate Change

Articles examining the impacts of climate change on country food access identified several ways that climate change challenges access to fish in Inuit Nunangat. Later and longer ice freeze-up has resulted in shorter fishing seasons and reduced predictability of the harvest season [34,59,73,74,75,76,81,84] Climate change also impacts the safety of harvesters while fishing on sea ice. For example, sea ice is recently reported to be thinner and weaker than it has been in the past [33,61,68]. Additional safety concerns that are linked to a changing climate include extreme and unpredictable weather and more frequent and intense winds [33]. Such challenges can inhibit the accessibility of fishing grounds and increase the risk of accidents while harvesting. Further, studies reported that climate change may be causing both positive and negative outcomes on fish, such as positive impacts on fish quality (condition factor) in Arctic Char and worsened health (e.g., parasites) and reduced flesh quality (e.g., changing for a paler colour) or nutrient content of Arctic Char and other country food species [59,61,65,68,69]. This was echoed by Inuit study participants in Cambridge Bay and Kugluktuk, Nunavut [59] and in Pangnirtung, Nunavut [61]. Relatedly, Gilbert and colleagues reported that Inuit in Cambridge Bay and Pond Inlet identified that the cumulative impacts of anthropogenic developments, such as mining, shipping, and tourism, also contribute to reduced access to fish and wildlife and low yield harvest [65].

#### 6.3.2. Socio-Economic Change

Inuit have undergone rapid social, cultural, and economic changes, largely driven by colonial government policies that have had wide-ranging impacts on fish harvesting. Fishing is often dependent on equipment such as motorboats, nets, and fishing rods. However, in the context of northern communities where incomes and economic opportunities are often limited, financial constraints were frequently cited as a barrier to fish harvesting [62,73,75,79]. In a qualitative study conducted in Kugluktuk and Cambridge Bay, Nunavut, Inuit expressed that access to fish was becoming more challenging due to socio-economic barriers, including erosion of sharing networks due to cultural change, reductions in active fishers in communities, and the increased monetization of country food [59,81]. Further, Inuit fish harvesters in Pond Inlet, Cambridge Bay and Pangnirtung, Nunavut, communicated that they have limited access to loans to purchase equipment and fishing gear [61,65]. Fish harvesters also face challenges related to obtaining commercial licenses and meeting processing requirements established by provincial and territorial governments to sell fish in the open market to non-Inuit [75,77]. For example, Gombay details the challenges of an Inuk fisher and entrepreneur, Malachi [63]. Malachi must overcome many barriers to sell his catch in the open market, including ensuring that the fish is transported according to food safety regulations (including ensuring they remain frozen in transit), managing the stigma of selling country foods, and navigating the threat that commercialization of country food could have on social relationships and community expression [63]. Finally, volatility in global fish markets impact the price and demand for fish and is often cited as a challenge to the implementation and stability of commercial fisheries, particularly small-scale ones [59,81].

#### 6.3.3. Sustainability and Stability of Fisheries and Fish-Related Community Programs

Two studies attempted to reconstruct and estimate the overall fish catches in the Canadian Arctic [34,59] and one evaluated the current and long-term potential of Arctic fisheries in Canada [85]. These studies used the five major large marine ecosystems (LMEs) Arctic boundaries as the study zones: (1) Canadian High-Arctic and North Greenland (formerly Arctic Archipelago) (CHA-NG), (2) Canadian Eastern Arctic and West Greenland (CEA-WG) (largely made up of Baffin Bay-Davis Strait), (3) Hudson’s Bay complex (HB), (4) Beaufort Sea (BS), and (5) Baffin Bay-Davis Strait (BB-DS). Below we summarize their key findings.

Driven predominantly by dog feed for sled teams, catches in the Arctic LMEs peaked in the 1960s, with Arctic Char accounting for most of the catch [33]. More recently, in the year 2000, Arctic marine harvests totalled 4882 (+/− 510) tonnes. 32% of this harvest was fish, with Arctic Char accounting for 22% (other marine harvested species included, Sculpin, Atlantic Mackerel, Whitefish, Salmon, Cod, Inconnu, Ciscos, Pacific Herring, Flounder, Shrimp, Turbot, ringed seal, bearded seal, harp seal, harbour seal, beluga, narwhale, walrus, and polar bear). Commercial fisheries harvesting Turbot, Shrimp and Atlantic mackerel accounted for 6% [82]. Between 2004 and 2014, average annual harvest across the Canadian High-Arctic and North Greenland LME, the Canadian Eastern Arctic and West Greenland LME, Hudson’s Bay complex LME, and the Beaufort Sea LME increased to 189,000 tonnes [85]. Arctic prawn (105,000 tonnes), Greenland Halibut (31,000 tonnes), and Atlantic cod (15,000 tonnes), dominate harvested species by weight, with 99% of total catches being harvested from the Canadian Eastern Arctic and West Greenland LME [70]. For subsistence purposes, fish were harvested in the BS LME the most (80%), while in the CEA-WG LME, subsistence catches comprised the least (0.5%) [85]. Although subsistence catches in the CEA-WG LME were the lowest by percentage, by weight they exceed total subsistence catch in the BS LME [85]. In the HB LME subsistence catches were significant at 717 tonnes, though were insignificant in the CHA-NG LME (<0.1 tonnes) [85].

Regarding current and future catch potential in the face of climate change, projected annual catch under a low climate change scenario was estimated to be similar to the current projected annual catch (4.16 million tonnes and $4.76 billion USD) at 4.16 (+/− 2.89) million tonnes and $4.76 (+/− 3.98) billion USD by the end of the century [85]. Under a high climate change scenario, catch and landed value increased to 6.95 (+/− 5.07) million tonnes and $8.45 (+/− 7.36) billion USD, by the end of the century [85]. Species with the greatest catch potential were capelin, beaked and golden red fishes, and Northern prawn [85]

The sustainability and stability of fisheries-related food security programs in Inuit Nunangat can pose challenges. In Nunavik, for example, there have been repeated attempts at establishing a community-based initiative to provide free Arctic Char to pregnant women [58]. Char was harvested by local fishers and sold to the Nunavik Regional Board of Health and Social Services (NRBHSS) for below market value. Char was then frozen and transported by air to communities with a little or no access to Arctic Char, and distributed by the nursing station in each community. The overall goal of this program was to increase access to nutritious and culturally desired food and decreased intake of contaminants [58]. This initiative was piloted in three communities in the early 2000s and there have been attempts to scale the program to include the Hudson Bay Coast [58]. In 2011, the NRBHSS began the initiative again [58]. Although this program has been well received, there were challenges in the reliability of transportation of the char to the community, timing of shipping to ensure freshness, and storage facilities [58]. Further, high human resource turnover rate and lack of communication and coordination among implementers posed challenges to the program’s fidelity [58]. Some stakeholders also expressed concerns over the procurement of char, a seasonal, migratory fish, and overexploiting the resource [58].

## 7. Discussion

In this review we characterized and summarized evidence in the published literature on the contribution of fish and seafood to food security in Inuit Nunangat and identified key themes as related to these topics. To date, most of the literature has focused on the safety of consuming marine harvested foods and the effects of climate change on fisheries. In Canada and globally, research on fish and seafood has clearly been identified as a pathway to improve food security and nutrition. However, in Inuit Nunangat, there is a paucity of research explicitly exploring the relative contribution and potential of fish and seafood as a means to achieve food security; this is especially true for the ISR and Nunatsiavut. Building on the results from this scoping review, we discuss three actions that may lead to research and policy that support the role of fish and seafood in food security in Inuit Nunangat: improved metrics, co-management of fisheries resources, and an integrated policy framework.

### 7.1. Improved Metrics on Fish and Seafood Abundance, Harvest, and Consumption

Over the course of this review, we identified that fish could contribute to food security directly through subsistence fishery access and fish consumption and indirectly through increasing household purchasing power through economic benefits fisheries provide and self-determination. However, very few studies have explored and elucidated the explicit pathways through which fish and seafood contribute to food security in Inuit Nunangat. Currently, a research gap exists in identifying the explicit associations between fish harvest and/or consumption and household food security. A gap also exists in studying the association between employment in the fish industry and household food security; this is especially true for crustaceans and molluscs. Additionally, while many studies note the importance of fish and seafood to the culture and diet of Inuit across Inuit Nunangat, including as an excellent source of many macro- and micronutrients, there are few studies that explore the estimated contribution of the consumption of commercial and subsistence fish in reducing adverse nutrition-related health outcomes. Future research should better document fish protein intake as a percentage of total protein intake as this gives critical information on the use and quality dimensions of food security; annual harvestable biomass per household which informs on the availability dimension of food security; as well as the contributions of harvested fish and income earned through commercial fishing to alleviating household food insecurity which informs on the accessibility dimension of food security. As commercial fisheries development increases there will be a need for improved harvest statistics and effective management tools to ensure the sustainability of important country food species, such as Arctic Char [88]. Further, the connectedness between subsistence and commercial fisheries and country food species (for example, the impact of commercial Halibut harvest on narwhal) remains largely unexplored. Ecosystem-based fishery management is not yet commonplace in the Arctic. However, researchers and decision-makers may benefit from employing an ecosystem-based fishery management framework to holistically understand the interactions between subsistence and commercial harvests within the marine system.

### 7.2. Improved Co-Management

Access and availability are two components of food security that can be directly influenced by policy and governance [89]. Many studies in this review described the importance of co-management and decentralization of fisheries management to enhance agency and decision making of Inuit community members. There remains an imbalance in power and a need to meaningfully include local perspectives in governance and decision-making of fisheries resources. As it pertains to fisheries resource management, DFO has the right to veto proposals by the joint management committees, however, Inuit do not have this same power. In the context of walrus harvesting and management, the Inuit Circumpolar Council recently reported “Inuit have stressed that true co-management would allow for Inuit representatives to hold substantial power in decision-making, including veto power” [25]. This power, by DFO, has also been exercised as recently as 2020, when the Minister vetoed a proposal by the NMRWB on beluga harvesting [90].

Meaningful co-management of fisheries requires a bottom-up approach to governance that enhances equity, improves local livelihoods, and facilitates access to fish as both a source of food and income [38]. This calls for more inclusive, innovative, and participatory governance of fisheries [91,92,93,94]. A successful bottom-up approach to governance will put citizens’ voices at the center of decision-making, enhance community-to-community and region-to-region relationships and communication in fisheries management, and encourage the conceptualization of fish as part of a larger social and natural ecosystem, a social-ecological system, rather than just as a resource [95].

### 7.3. Integrated Policy Framework

Some researchers have recently developed and promoted the “fish as food” food systems approach, which may be a promising framework to examine and understand the relationships between fisheries (commercial and subsistence) production and consumption, while also recognizing the environmental, social, and economic dimensions that affect food and nutrition outcomes [95,96,97,98,99,100]. This framework provides researchers and policy makers the opportunity to view fisheries and food policy together rather than in silos [99,101,102]. This framework would benefit from including other marine and freshwater species such as seafood but also marine mammals, which play a key role in Inuit culture and diet. It will also require new areas of research that transcend traditional disciplinary boundaries, combining social science, food systems, health policy, ecology, and social justice research. Further, this will require not only collaboration across academic disciplines and government sectors (including Inuit colleagues) but will also meaningful collaboration with different knowledge systems and perspectives [98,102,103].

In June 2019, the Canadian federal government passed Bill C-68, which introduced amendments to the federal *Fisheries Act*, the main statute that regulates fisheries and aquaculture. These amendments promoted the importance of Indigenous knowledge and social, economic, and cultural considerations in Canadian fisheries [30,104]. This amendment lays the groundwork to appreciate the responsibility of fisheries regulations to consider the importance of fish to food security in Inuit communities, however, most federal, and provincial policies in Canada remain focused on fish as a commodity for its economic value and exportation potential [105]. For example, the Aboriginal Fisheries Act has no mention of health-related themes [106]. Conversely, Nutrition for Health, a national policy and strategy, has no mention of fisheries-related themes [106].

Related to governance and decision-making power, our review revealed a need for an integrated fisheries policy that views fish and seafood as more than a commodity, but also an integral part of food systems in northern regions of Canada. Despite the importance of fisheries to health and food security, social development and human health are not often prioritized within fisheries management practices [91,95,96,99,103,107,108,109]. Further, food security research often focuses primarily on the affordability of food and neglects the important contributions of country food systems to diets and subsistence economies [31,110]. Effectively, fisheries policies are at the intersection of food, economic, and environmental processes and should aim to contribute to food security, economic growth, social justice, and environmental sustainability in equal measure. Therefore, fisheries policy cannot be considered successful until it has demonstrated an improvement in food and nutrition security [98].

## 8. Limitations

This review was subject to several limitations. First, we restricted our search to articles available in English, which may have excluded relevant information and research published in other languages. Second, due to the multidisciplinary nature of fisheries and food security research, there was substantial heterogeneity of study characteristics and design within included studies, which created challenges to synthesizing and comparing published literature. However, a strength of the scoping review methodology is its ability to meaningfully incorporate research from varying disciplines and study designs. Finally, a substantial limitation is that this review excluded grey literature, which may have resulted in an exclusion of relevant community initiatives and policy documents. In particular, there is a robust catalogue of DFO, provincial, and regional fisheries management reports that summarize stock assessments, fisheries policies, and fisheries harvest data.

## 9. Conclusions

This review affirms that fisheries play an important role in food security in Inuit Nunangat. We found that while pathways connecting fisheries to food security were explored in peer-reviewed literature, there remains a need to better understand the explicit associations between fisheries and household food security. With the recent rise in academic literature exploring fish and food security, there is a growing recognition of the importance of fisheries in alleviating food insecurity in Inuit Nunangat. Going forward, there is a need for improved metrics on seafood contribution to food security, fish and seafood abundance, harvest, and consumption as it relates to food security. Additionally, researchers should further explore how the needs and responsibilities of fisheries can be balanced with the needs of local communities to promote Inuit culture and social justice, and to enhance food, nutrition, and livelihood security [99].

With the likely rise in commercial fishing operations due to the increased access to the Arctic, it is crucial that this enterprise is informed by, and inclusive of, Indigenous knowledge and practices [93]. Given the possibility that commercialization could reduce access and availability of fish to the community, it becomes even more imperative that the needs of local community food security are balanced with the needs of the commercial enterprise through a leading role in the governance and management of the fisheries resources [109].

## Figures and Tables

**Figure 1 ijerph-20-02629-f001:**
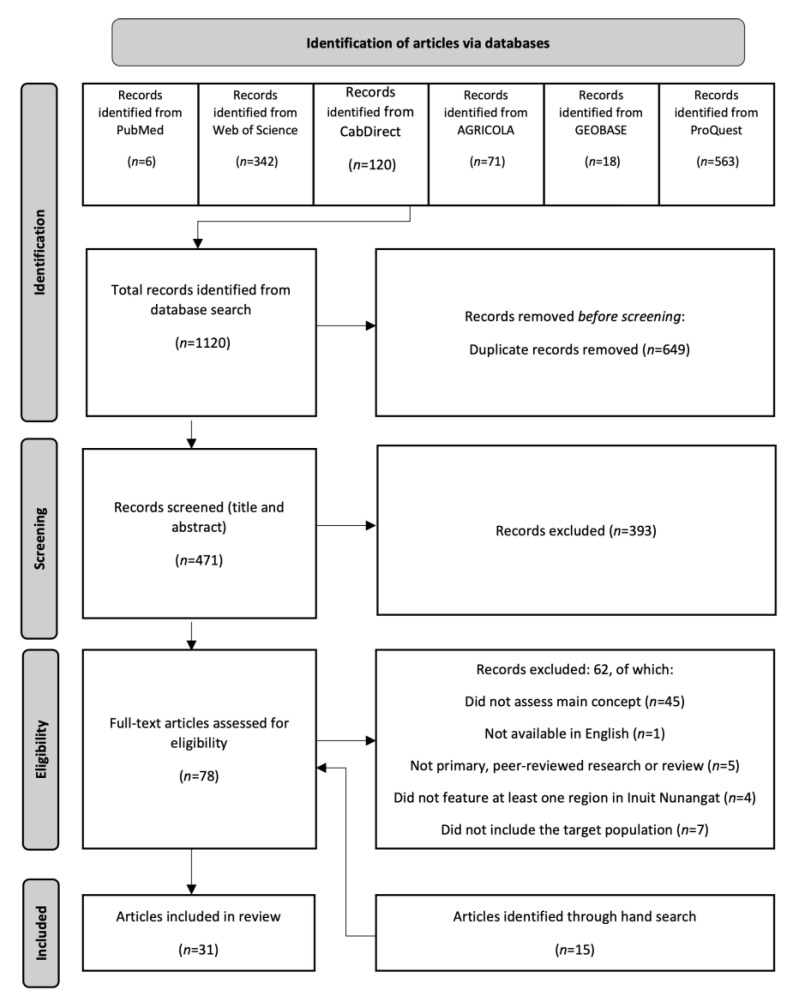
**PRISMA** flow diagram for identifying articles on the role of fisheries in food security in Inuit Nunangat.

**Table 1 ijerph-20-02629-t001:** Example search terms used in database searches to identify studies on the role of the fish and seafood on food security in Inuit Nunangat (* indicates a truncation Boolean operator).

Main Term	Expanded Term
Food Security	food security OR food supply OR nutrition security OR food system OR food sovereignty OR food availability OR food accessibility OR food utilization OR traditional food OR country food OR food preference OR diet OR consumption
Fish	Fish * OR aquaculture OR seafood OR marine harvest
Arctic	Arctic OR Arctic Canada OR Canadian OR Canadian Arctic OR Nunavut OR Inuvialuit OR Nunavik OR Nunatsiavut OR Inuit Nunangat OR Yukon
Inuit	Inuit OR Indigenous OR ^†^ Eskim * OR Aboriginal

^†^ In order to develop a search string that would capture all articles potentially relevant to Inuit, it was necessary to recognize the unethical terminologies that have been (and occasionally continue to be) used to describe Indigenous Peoples in academic research [54,55]. At least one of the search terms used in this list is no longer considered appropriate for use with Inuit communities, and is frequently considered offensive; it is a relic of colonial academic scholarship and was included to capture literature from decades prior to the discontinuation of the term. Though the authors have included this term, it in no way reflects their beliefs and opinions about, or relationships with, Indigenous Peoples.

## Data Availability

Available upon request.

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
