# Peer review of "Identifying Barriers and Pathways Linking Fish and Seafood to Food Security in Inuit Nunangat: A Scoping Review"

_ijerph, 2023, doi:10.3390/ijerph20032629_

Round 1

Reviewer 1 Report

This manuscript explores the range, nature, and extent of literature examining the role of fish in food security in Inuit Nunangat, Canada. Specific objectives are to understand: 1) what topics and themes have been widely explored in the literature related to barriers and pathways linking fish and seafood to food security; 2) where research, policy, and action gaps exist; and 3) how fisheries currently contribute to food security. This is an important topic, and the study deserves a place in the journal, but this manuscript needs some work before publication.

My comments are as follows.

Introduction

L113: Indicate this value CAN$86.3 related to which year or time.

L181-182: Same as above, please add the year or time

Materials and Methods 

L188: fix citation... not et al. it Arksey and O’Malley

L234-236: This is confusing to me. How do you calculate this, and what do that numbers mean here

Results

L252-254: I would delete this. I don't know how it adds value to the section. 

L255-258: Please move this figure to the Methods section. 

Figure 1: All numbers added perfectly except the Articles identified through hand search. Please modify for clarity. 

You can start the results section with Study characteristics.

Table 2: This is too long and difficult to follow. I would move this to Supplementary Materials. I suggest adding a high-level summary table instead. This table could consist of columns such as pathways, key themes that emerged, examples of regions/communities/fisheries, and relevant citations from the 30 studies.

Figure 2: This figure needs some modifications and isn't easy to understand. What does Thematic Analysis do in the Results section—or is this part of the Methods? What do these different arrows mean? What are these two boxes? It's not readable in my black-and-white printout. Also, this might not fit well in this section. Most importantly, you need to explain this figure in the main text.

L300: "store-bought fish across Inuit Nunangat" I know what you mean here, but this can be misleading to some readers. I know many Inuit who buys "wild caught" Arctic Char from local stores because sometimes local stores buy fish from local fishers.

   **Citation style: There are many citation errors and inconsistencies to fix throughout the manuscript. (e.g., L341,353,371,373, 399,401, 453).

L415-442: I would split this long paragraph into two.

L457-458: What does "ex." means here?

Overall, I enjoyed reading this paper, and I believe it deserves a place in IJERPH with revisions. 

Reviewer 2 Report

Good review paper. Minor editorial errors. Mature reasoning.

1. What is the main question addressed by the research?
This review affirms that fisheries play an important role in food security in Inuit Nunangat.

2. Do you consider the topic original or relevant in the field, and if
so, why?
The theme is as original as possible. None of the known researchers takes such an approach to teamtics. The paper concerns the narrow community of Nunangat.

3. What does it add to the subject area compared with other published
material?
 With the recent rise in academic literature exploring fish and food security, there is a growing recognition of the importance of fisheries in alleviating food insecurity in Inuit Nunangat.
Additionally, researchers should further explore how the needs and responsibilities of fisheries can be balanced with the needs of local communities to promote Inuit culture and social justice, and to enhance food, nutrition, and livelihood security. The paper is a comprehensive overview.

4. What specific improvements could the authors consider regarding the
methodology?

The authors could supplement the paper with a life cycle method.

5. Are the conclusions consistent with the evidence and arguments
presented and do they address the main question posed?

Yes. The conclusions are correct.

6. Are the references appropriate?
Yes

7. Please include any additional comments on the tables and figures.

Some Figures are inadequate quality.

Reviewer 3 Report

Generally, I suggest using the 3 questions on p. 2 (lines 74-77) as research questions instead of the question on p. 5 (lines 195-196) checking the key findings in table 2 in accordance with them and resuming answers in the conclusion.

Afterwords I suggest to review the abstract to make it more effective.

The possibility that commercialization could reduce access and availability of fish to the community seems an important argument: Is it possible to analyse it more in depth?

Specific suggestions:

lines 20-21: specify kind of study designs, methods, and methodologies. 

line 186: what is "they" referred to?

lines 198-203: key concepts and adaptation for each database are not present in table 1. More correspondence between text and table is needed 

lines 252-254: use present tense

Figure 1: insert references to supplementary material to understand exclusion criteria

line 647: insert extended wording for DFO

Round 2

Reviewer 1 Report

The authors have addressed my comments. I don’t have any comments or recommendations for this manuscript. Co-authors: All the best for your hard work!